# Produce or Buy: Impacts of Citrus Farming and Crop Diversification on Household Dietary Diversity in Guangxi, China

**Xinjian Chen *** , **Baoji Zhou, Xinfeng Zuo and Xiaojun Fan**

School of Economics, Guangxi University, 100 Daxue Road, Nanning 530004, China;
zhoubaoji3154@163.com (B.Z.); xfzuo@st.gxu.edu.cn (X.Z.); gxufxj@gxu.edu.cn (X.F.)
* Correspondence: xjchen@gxu.edu.cn

**Abstract:** In recent decades, China has made significant strides in food and nutrition security, yet challenges persist in the western rural regions, especially in Guangxi. Farming households in this area face heightened vulnerability due to limited arable land and environmental hazards, leading to increased efforts to boost income through horticultural crop farming. This study explores the impact of citrus farming and crop diversification on household dietary diversity within the context of the existing literature, examining trade-offs between subsistence and income-generating farming. Using OLS regression techniques and a mediating effect model, the analysis focuses on distinct contexts within citrus farming, utilizing survey data from households engaged in the cultivation of both citrus and non-citrus. Results reveal that while citrus cultivation moderately contributes to dietary diversity, its primary influence lies in encouraging farmers to diversify food purchases, with the mediating effect from market purchases exceeding 50%. Diversifying crop production, including staple crops and vegetables alongside citrus, proves more effective in enhancing dietary diversity among citrus farming households. Crop diversification positively influences dietary diversity, partially addressing household self-sufficiency. While extensive crop diversification may not be the ultimate solution to food security challenges, promoting specific diversification systems shows promise in the context of sustainable agriculture goals.

**Keywords:** citrus cultivation; crop diversity; food security; dietary diversity; sustainable agriculture





## 1. Introduction

In the face of remarkable demographics, China sustains almost one-fifth of the global population, yet it does so with a meager 9% of the world's arable land and just 6% of its freshwater resources. This paradox of population and resources underscores a tight balance between food supply and demand in China [1,2]. Over the past decades, the nation has undergone transformative changes catalyzed by policies such as reform and opening up, a great western development strategy, and targeted poverty alleviation. These measures have allowed China's rural areas to bid farewell to a history marked by food insufficiency, resulting in substantial improvements in food and nutrition security [3]. However, it is worth acknowledging that the specter of food and nutrition insufficiency still looms large in the less-developed rural regions of western China. Malnutrition extends beyond mere food quantity or calorie intake. The quality of the diet, particularly the diversity of food groups consumed, holds paramount significance [4]. Western regions, like Guangxi, not only struggled to provide basic food provisions but also grappled with severe issues surrounding food and nutrition security [5,6]. Thus, while food security remains imperative, there is a persistent need to intensify efforts aimed at improving the quality and diversity of diets among the entire rural population. This persistent challenge looms large as a major concern [7].

In recent years, farmers in Guangxi, China, have undertaken persistent adjustments to their production and planting structures, driven by the goals of poverty alleviation, income augmentation, and the resolution of fundamental food security concerns. This transformative process has resulted in a growing number of households engaging in citrus cultivation [8]. Notably, citrus cultivation surpasses cereal crops in comparative returns. It has emerged as a pivotal agricultural industry in rural Guangxi, playing a crucial role in facilitating poverty alleviation and income enhancement for farmers [9]. The expanding cultivation area dedicated to citrus at the household level is anticipated to lead to a reduction in the acreage allocated for cereal crops and other plants. Consequently, this shift is expected to disrupt the longstanding self-sufficiency in food consumption patterns among farmers, ushering in a discernible trend toward procuring food from the market [10]. Farmers in the rural regions of western China are predominantly characterized by small-scale farming practices, with the majority managing land holdings of one hectare or less. The evolution in citrus cultivation practices prompts an inquiry into whether it has effectively contributed to enhancing food and nutritional security for small-scale farming households. The source of diversity in household food consumption—whether primarily derived from market-driven factors or arising from diversified production for self-sufficiency within farming households—remains an unresolved aspect.

A substantial body of research on agriculture and nutrition health underscores the positive impact of cultivating cash crops and embracing crop diversification on smallholder nutrition health. Vegetables and fruits are the most cost-effective sources of vitamins and minerals, playing an indispensable role in maintaining good health [11]. For smallholder farmers in developing countries, fruit consumption is often limited in their daily diets. Thus, cultivating fruit trees on their own can significantly boost their daily fruit intake [12]. The cultivation of high-profit economic crops like horticultural fruits not only increases farmers' income [13,14] but also enhances their financial capacity for staple crop cultivation [15]. Consequently, this contributes to elevating their food and nutrition security levels, facilitating broader food purchases from the market [16,17]. However, it is essential to acknowledge that certain scholars have identified an inverse relationship between smallholder food security and the cultivation of cash crops [18]. This relationship is attributed to the fact that the cultivation of non-food cash crops can encroach on the allocation of land for staple crops and other essential agricultural produce [19]. Furthermore, the diversity of crops on a farm is a pivotal factor in augmenting the food and nutrition security of smallholders [20]. Crop diversification on a farm is commonly perceived as a strategy that enriches the quality and diversity of smallholders' diets [21]. In most scenarios, increased diversity in farm production serves as a welfare-enhancing strategy [22], exerting a favorable influence on food security and nutrition for households in low- and middle-income countries [23–26]. Nevertheless, it is crucial to note that different scholars have arrived at varying conclusions, suggesting that improving smallholder access to markets may yield more effective results in terms of nutritional improvements than promoting diversity in self-sustaining farm production [10,27,28].

The significance of specialized horticultural fruit tree cultivation and crop diversification in the context of smallholder households' nutrition and health remains a focal point of inquiry. Amid the ongoing transformation in horticultural agriculture, characterized by trends toward specialization and moderate scaling, the pertinent question arises: do smallholder farmers continue to depend on crop diversification as a means to enhance food provision and nutrition within their households? Moreover, given that Guangxi stands as China's foremost fruit-producing province, it underscores the need to examine whether smallholder households are affected by their engagement in fruit tree crop cultivation in terms of food and nutrition security. Consequently, this area warrants further research and exploration. At present, a definitive consensus on the relationship between farm crop diversity and household dietary quality remains elusive, with most studies failing to deliver consistent findings, particularly concerning distinctions based on regional or production-type variables [29]. However, specific insights into regional and crop-type contexts play a

pivotal role in shaping policies that aim to improve food and nutrition health outcomes. While several studies have delved into rural livelihoods and food nutrition challenges in China, particularly within the relatively impoverished western and central regions [30], rigorous analyses grounded in household survey datasets on this subject remain relatively scarce. Research on smallholder households engaged in horticultural fruit tree cultivation is even scarcer.

In response to this research gap, this article employed a series of multiple linear regression models to examine the changes in food and nutrition consumption among citrus growing households, using household survey data from Guangxi, China. The primary objective of the empirical analysis was to scrutinize and compare the impact of citrus farming and crop diversification within households on smallholder food and nutrition health outcomes, assessed through the Household Food Consumption Score (FCS). We investigated two distinct livelihood backgrounds: smallholder farmers involved in citrus cultivation and those not participating in citrus cultivation. In the agricultural system of citrus farming, the scope for crop diversification is somewhat limited due to the prevalent high levels of specialization among farmers, which might potentially result in a reduced overall number of different crops being cultivated. Additionally, the analysis was designed to acknowledge the role of crop diversification as it relates to the market, self-sufficiency, or a combination of both. This approach is intended to enhance our comprehension of the intricate balance within households concerning self-sustaining agriculture versus more profitable horticultural agriculture in the context of food and nutrition security. Specifically, this article seeks to investigate whether disparities exist in the food and nutrition contributions of crop diversification between households engaged in citrus cultivation and those not involved in such practices. This inquiry is prompted by the fact that crop diversification has been explored as an adaptive strategy for market-oriented and self-sufficient smallholder agricultural households [16].

## 2. Food Security and Citrus Industry in Guangxi

According to the World Bank's poverty threshold, which stipulates a daily living expense of USD 1.9 [31], Guangxi marked a significant milestone by achieving full poverty alleviation in 2020. Nonetheless, persistent nutritional challenges exist, particularly in rural areas of Guangxi. Analysis of nutritional survey data conducted by the Guangxi People's Government during the period of 2015–2017 reveals a prevalence of overweight in individuals aged 18 and above, standing at 30.4%. Among children under the age of 2, the rates of stunted growth and anemia are notably high, at 8.9% and 39.6%, respectively, while primary school students exhibit a malnutrition rate of 16.9%. Furthermore, a substantial portion of the population grapples with issues such as micronutrient deficiencies and inadequate dietary fiber intake. Over the long term, Chinese residents have exhibited a propensity for excessive consumption of cooking oil and salt while falling short in their consumption of whole grains, dark-colored vegetables, fruits, dairy products, fish, and shellfish, as well as leguminous foods. The per capita daily intake of whole grains and leguminous foods among adults hovers at less than 15 g, with over 80% of the adult population experiencing a severe deficit in their dietary intake [32].

The Chinese government places particular emphasis on food security and has enacted a series of policies to ensure self-sufficiency in staple crops. Since the onset of the 21st century, China has consistently witnessed an increase in cereal production, with a per capita cereal availability of 486.1 kg in 2022, surpassing the internationally recognized food security threshold of 400 kg. Overall, China has achieved food security. However, Guangxi, located in the economically disadvantaged western region characterized by mountainous terrain and limited arable land, lags behind in economic development, with a per capita cereal production of only 275 kg. To bolster rural residents' incomes and mitigate the threats to food and nutritional security, promoting and supporting the cultivation of high-yield horticultural and fruit crops on non-cereal lands has emerged as a pivotal strategy to ensure food and nutrition security. Against this backdrop, Guangxi has ascended to be the leading

province in China for fruit production, ranking among the top ten provinces in vegetable production and holding the top position in citrus fruit production. In 2022, the fruit yield of Guangxi province exceeded 31 million tons, while vegetable production exceeded 40 million tons (Figure 1). The extensive cultivation of horticultural crops, especially the large-scale expansion of fruits, has significantly alleviated the problem of insufficient food consumption in rural areas of Guangxi.

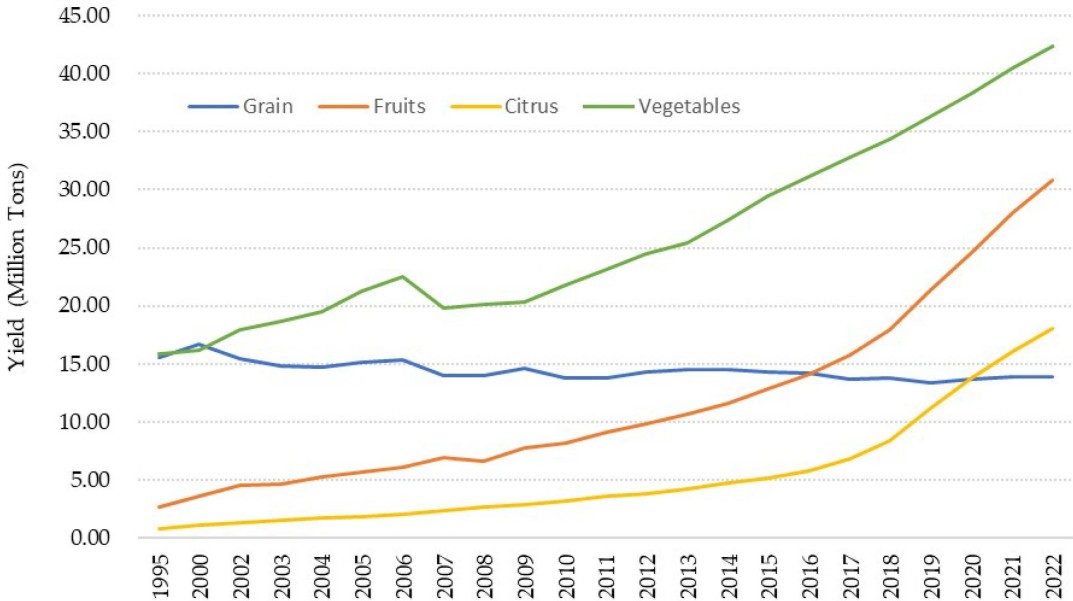

**Figure 1.** Grain, vegetables, fruits, and citrus yield in Guangxi (1995–2022). Source: own representation, based on data from the Guangxi Statistical Yearbook 1996–2023 [33].

The citrus industry plays a pivotal role in Guangxi's agricultural economy. In 2022, the citrus cultivation area in Guangxi spanned 9.46 million mu (equivalent to approximately 630,000 hectares), yielding a harvest of over 18 million tons of citrus valued at USD 8 billion. This contribution constitutes 12% of Guangxi's total agricultural production value. Citrus occupies a central position within the realm of agricultural commodities, playing a dual role by contributing to rural poverty alleviation and income augmentation in Guangxi while also serving as a substantial fruit product destined for both China's eastern coastal regions and export markets abroad. Figure 2 illustrates the regional distribution of citrus production in Guangxi, emphasizing the widespread presence of citrus cultivation throughout the entire province. Guilin emerges as the leading region with the largest citrus cultivation area, yielding an impressive 6.9 million tons. Other major citrus-producing areas in Guangxi include Nanning, Hezhou, Laibin, and Liuzhou, all boasting annual citrus yields surpassing 1 million tons. Our farming household survey data reveals that, on average, households cultivating citrus manage an orchard of 4.3 mu (approximately 0.29 hectares) and enjoy an annual per-household income exceeding RMB 30,000. This substantial boost significantly bolsters rural household incomes. However, the continual expansion of citrus cultivation may come at the expense of cereal crops and agricultural diversity. Citrus, as a perennial crop, is unsuitable for the simultaneous cultivation of other crop types through intercropping or crop rotation. Consequently, while smallholder citrus farming elevates household income and encourages market-based food purchases, it may curtail household self-sufficiency in food, thereby impacting household food security and nutritional adequacy. In the forthcoming sections, we will elucidate our chosen research sites and employed methodologies, aiming to gain a more profound insight into crop diversification and the repercussions of citrus cultivation on the food and nutritional security of smallholder households.

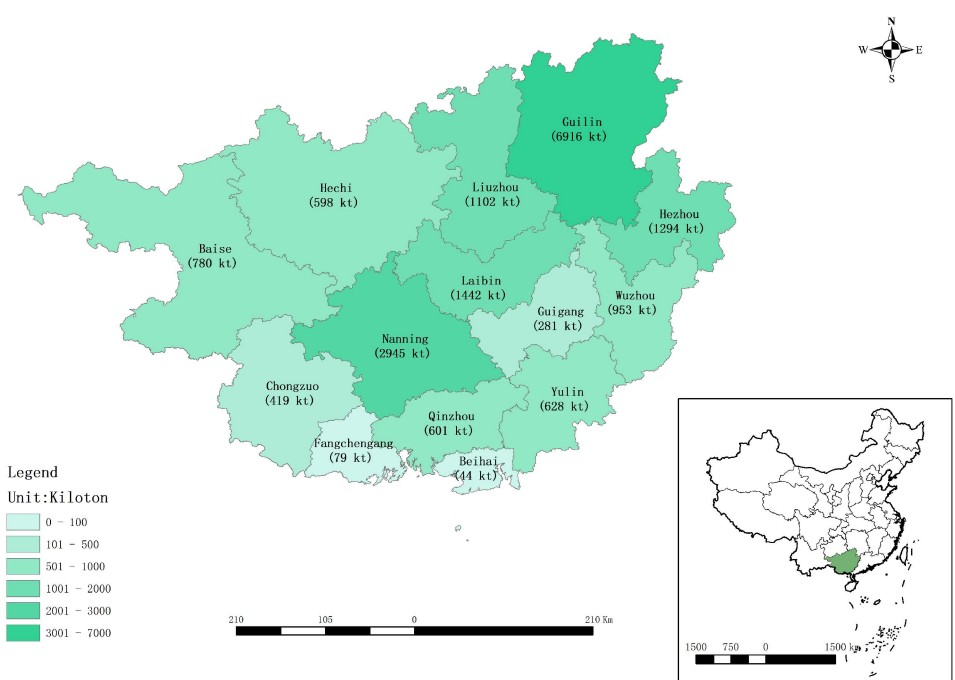

**Figure 2.** Distribution of citrus production in Guangxi (2022). Source: own representation, based on data from the Guangxi Fruit Production Technology Guidance Station [34].

## 3. Materials and Methods

Based on the current food safety situation in Guangxi, China, and the trend of citrus farming, this study aims to investigate the food and nutritional security changes brought about by citrus cultivation among farmers. Additionally, it seeks to assess whether citrus-growing households and those not engaged in citrus cultivation continue to maintain the necessary crop diversity to provide diversified food for their families. Therefore, drawing on existing research literature, this article proposes the following research hypotheses: crop diversity enhances dietary diversity for household farmers in both citrus-growing and non-citrus-growing households; citrus cultivation is posited to prompt households to augment dietary diversity through the procurement of food from the market.

### 3.1. Household Survey and Data Collection

A standardized household survey employing face-to-face interviews was undertaken within the citrus-producing region of Guangxi, China. The survey spanned a duration of six months, commencing in July 2021 and concluding in March 2022, with additional data collection conducted in the latter stages of the survey. The survey encompassed two distinct categories of households: those actively involved in citrus cultivation within the primary citrus-producing areas of Guangxi and those not engaged in citrus cultivation. The household survey of this study covered seven major citrus cultivation regions in Guangxi, which collectively contribute to more than 80% of the province's citrus production. These regions include Guilin, Nanning, Hezhou, Laibin, Liuzhou, Baise, and Wuzhou, as visualized in Figure 2. To gather data, the survey employed a stratified random sampling approach, stratifying based on the distribution of citrus cultivation acreage at the county level. In total, a representative sample of 720 households was selected, hailing from 48 villages situated within 24 townships across 12 counties in the aforementioned seven cities. Of these households, 480 were actively involved in citrus cultivation, while the remaining 240 were not.

To implement our sampling strategy, we initially prioritized the selection of the top 12 counties based on citrus production among the seven cities to ensure a comprehensive representation of citrus production intensities. Following this, two townships were ran-

domly chosen from each of these counties, followed by the random selection of two villages within each selected township. Subsequently, we randomly sampled 15 households from each village's household roster, comprising 10 households engaged in citrus cultivation and 5 households not involved in citrus cultivation. Out of the 720 households initially sampled, 429 households engaged in citrus cultivation and 213 households not involved in citrus cultivation successfully participated in our survey. The remaining households were unable to be reached due to factors such as their absence or unavailability for contact. Given that this study's primary focus is to investigate the impact of smallholder citrus cultivation and crop diversity on dietary diversity and to maintain a balanced representation of households engaged in citrus cultivation and those that are not, we excluded specialized citrus cultivation households from our dataset. These specialized citrus cultivation households exclusively grow citrus crops, refrain from cultivating other crops, and rely solely on market purchases for their daily food consumption. Consequently, our final dataset comprises 262 households engaged in citrus cultivation (who also cultivate other crops) and 213 households not involved in citrus cultivation, resulting in a total of 475 cases available for analysis.

### 3.2. Dietary Diversity Indicators

In this study, the dependent variables are farmers' food security and dietary diversity. Dietary diversity is commonly used to represent the quality of farmers' diets and is a readily measurable indicator of their daily nutritional needs. At the household level, a widely utilized variable is the household Food Consumption Score developed by the World Food Programme (WFP). FCS is a widely employed methodology in appraising food security, assessing the nutritional quality of household diets by examining the diversity and frequency of various food groups consumed over a specified timeframe [35]. Assigning scores to distinct food items based on their nutritional relevance, the FCS aggregates these scores to provide a comprehensive measure of dietary diversity. This approach aids in evaluating the nutritional well-being of populations, scrutinizing staple foods, fruits, vegetables, and animal products within their diets. FCS is computed based on the frequency of household consumption of diverse food categories over the preceding seven days, serving as a tool to evaluate food and nutrition security specifically within farming households. During the farmer household surveys conducted in this study, information was obtained regarding the frequency of consumption of nine different food categories over an average week, representing the typical components used in the calculation of FCS. These food categories include main staples such as rice, pasta, millet, maize, potato, yam, cassava, and other cereals, categorized as cereals; meats such as pork, fish, eggs, goat, beef, chicken, and seafood, grouped as the meat category. Additionally, data on legumes, milk, oil, fruits, vegetables, sugar, and condiments were collected. In the FCS calculation, food categories were assigned the following weights: main staples (2), legumes (3), vegetables (1), fruits (1), meat and fish (4), milk (4), sugar (0.5), oil (0.5), and condiments (0). Frequencies were measured as never (0), hardly at all/one day per week (1), every second to three days (3), most days (5), and daily (7). Based on the FCS calculation methodology described above, we summarized the food consumption score results and distribution for citrus-growing farmers and non-citrus-growing farmers, as illustrated in Table 1. Using widely accepted standards, the food consumption of households was categorized into four groups: Poor consumption (FSC of 28 and lower), Borderline food consumption (>28 and below 42), Acceptable low (42–52), and Acceptable high (>52). Based on the descriptive statistics presented in Table 1, a predominant number of households attained a rating at the "Acceptable High" level. The average FCS for households engaged in citrus cultivation is 63.44, whereas for those not involved in citrus cultivation, the average FCS is 56.25. Interestingly, citrus-growing farmers exhibited significantly higher FCS scores than non-citrus-growing farmers.

**Table 1.** The mean difference of FCS by food group: citrus farming and non-citrus farming households.

| Food Group | Foods and Nutrition | Citrus Farming Households ($n$ = 262) | Non-Citrus Farming Households ($n$ = 213) |
|---|---|---|---|
| FCS | Household Food Consumption Score | 63.44 ** (11.57) | 56.25 (12.61) |
| Cereals | Cereals, grains, roots, and tubers, such as rice, pasta, bread, sorghum, millet, maize, potato, yam, cassava, etc. | 12.94 (1.51) | 13.12 (1.17) |
| Pulses and nuts | Pulses/legumes, nuts, and seeds, such as beans, soy, pigeon peas, peanuts, or other nuts. | 9.05 ** (6.69) | 12.54 (5.99) |
| Milk and dairy | Milk and other dairy products, such as milk, yogurt, and other dairy products. | 8.06 ** (4.01) | 5.03 (4.82) |
| Meat, eggs, and fish | Meat, fish, and eggs, such as goat, beef, chicken, pork, fish, eggs, seafood, etc. | 19.69 ** (6.17) | 12.95 (9.86) |
| Vegetables | Vegetables and leaves, such as spinach, onions, tomatoes, carrots, peppers, green beans, lettuce, leafy greens, etc. | 5.05 ** (1.41) | 5.61 (1.32) |
| Fruits | Fruits, such as bananas, apples, lemons, mangoes, peaches, etc. | 4.07 ** (2.27) | 2.44 (1.91) |
| Oil and fat | Oil/fat/butter, such as vegetable oil, palm oil, shea butter, margarine, and other fats/oil. | 3.48 (0.08) | 3.47 (0.09) |
| Sugar | Sugar, or sweet, such as sugar, honey, jam, candy, cookies, pastries, cakes, sugary drinks, and other sweets. | 1.12 (0.95) | 1.08 (0.92) |

Notes: Standard deviations are reported in parentheses. ** indicates the variable mean differs from that of non-citrus farming households used in $t$-tests at 1% levels. Source: WFP and the authors' survey.

### 3.3. Crop Diversification Indicators and Independent Variables

This study uses the Shannon index to measure a farmer's household's crop diversity at the farm level, and a higher index indicates greater crop diversity. In the domain of agricultural ecology, the Shannon index is frequently utilized to gauge the variety of crops within a specific ecosystem, specifically focusing on the abundance and uniform distribution of crop groups within the agricultural landscape [36]. The computation entails the application of the Shannon index to these crop groups, taking into consideration the mere enumeration of distinct crop groups cultivated within the preceding 12 months. Assuming that the count of farmer $i's$ planted crops is $N_i$, the Shannon index for farmer $i$ is calculated as follows [37]:

$$D_i = -\sum_{n_i=1}^{N_i} landshare_{ni} \ln(landshare_{ni}) \qquad (1)$$

where $D_i$ is Shannon index for farmer $i$, and $landshare_{ni}$ is the share of the farmland area cultivated with crop group $n_i$. Hence, the Shannon index accounts for the evenness of land allocation to different crop groups. The cultivation of citrus crops and crop diversity among farmers constitute two pivotal explanatory variables under investigation in this study. In addition to these two variables, we have also incorporated relevant control variables based on a comprehensive review of existing literature [2,38], as itemized in Table 2. Control variables encompass general demographic information, socioeconomic factors, and the share of food purchased from the market. Demographic variables include the age of the household head, ethnic minority status, educational attainment of the household head, and the size of the farming household [8,13,15]. The socioeconomic factors of the farming household encompass fixed assets, landholding size, non-agricultural income, and per capita household expenditure [24,29]. Furthermore, independent control variables encompass the degree of market dependency in terms of food consumption, denoting the proportion of all food items consumed within the household that are acquired through market transactions [24,38]. The basic characteristics of the interviewees and the analyzed sample are elucidated through the descriptive statistics presented in Table 2. The average age of the sampled households is 49 years, with an average educational attainment of

7.7 years, equivalent to an educational background spanning from primary to junior school. These findings signify a demographic profile characterized by aging and lower educational levels among the surveyed households. The average farmland size is 0.56 hectares, further underscoring the small-scale nature of the sampled farming households. Notably, 41.7% of households' food supply is acquired through market purchases, indicating a simultaneous reliance on market sources while retaining a substantial degree of self-sufficiency.

**Table 2.** Descriptive statistics of independent variables.

| Variables | Definition | Mean | SD | Min | Max |
|---|---|---|---|---|---|
| Citrus farming | 1 if household farming Citrus, 0 otherwise | 0.551 | 0.497 | 0 | 1 |
| Crop diversification | Shannon's Diversification Index | 1.181 | 0.349 | 0.23 | 1.76 |
| Age | Age of household head (years) | 49.021 | 9.201 | 25 | 69 |
| Edu | Education of household head (years) | 7.726 | 2.266 | 2 | 15 |
| Farmland | Total household farming land (hectares) | 0.567 | 0.455 | 0.1 | 4.2 |
| Off-farm | Share of family off-farm income (%) | 0.235 | 0.227 | 0 | 0.86 |
| Com. Food level | Share of all consumed food purchased on the market (%) | 0.417 | 0.173 | 0.16 | 0.75 |
| Per capita exp. | Household per capita expenditure (RMB 1000/month) | 1.414 | 0.591 | 0.42 | 2.5 |
| Family size | Household size | 4.176 | 1.327 | 2 | 8 |
| Ethnicity | 1 if the household's ethnicity is minority, 0 otherwise | 0.364 | 0.481 | 0 | 1 |
| Fixed asset | 1 if the household owns cars, 0 otherwise | 0.265 | 0.441 | 0 | 1 |
| Observations | | 475 | | | |

Source: authors' survey.

*3.4. Estimation Methodology*

To investigate the relative impacts of citrus cultivation and crop diversity on household food and nutritional security, herein referred to as FCS, we developed a series of multiple linear regression models with farming households as the units of analysis. Additionally, given the evident disparities in descriptive statistics of food consumption scores between households that engage in citrus cultivation and those that do not (see Table 1), we computed the interaction effect between the variable "citrus farming" and the independent variable "crop diversity". The use of interaction terms is a widely employed approach for assessing how explanatory variables differentially affect the outcome variable between two distinct groups [39,40]. They aid in understanding how the influence of crop diversity on food consumption scores varies depending on whether households engage in citrus cultivation or not. The resulting multivariate linear regression model is represented as follows:

$$Y_i = \beta_1 \, Citrus_i + \beta_2 \, D_i + \beta_3 X_i + \beta_4 Citrus_i \times D_i + \varepsilon_i. \tag{2}$$

In Equation (2), the dependent variable, denoted as $Y_i$, represents the estimated FCS of farmer households. Household farming citrus, denoted as $Citrus_i$, takes a dichotomous value of either one or zero. The level of crop diversity among households is denoted as $D_i$, while $X_i$ is a set of demographic and socioeconomic characteristics that may influence a household's dietary diversity. The vectors of parameters to be estimated are denoted as $\beta_1$, $\beta_2$, $\beta_3$, and $\beta_4$. The normally distributed random disturbance is denoted as $\varepsilon_i$.

In the empirical analysis, this article fitted several linear regression models in a stepwise fashion. In the first step, the model included only the variable of citrus farming to estimate the statistical differences in FCS between households engaged in citrus cultivation and those that were not. In the second step, the study further introduced crop diversity and the remaining control variables. In the third step, the model additionally incorporated interaction effects, capturing the association between citrus cultivation and crop diversity, on the foundation of the second step. In the fourth step, for robustness checks and to explore the differential impacts of households in distinct groups on food consumption, the article used group food consumption scores for each food category as the dependent variable and conducted further regression analyses following the three aforementioned models. Lastly, this study further employs Baron and Kenny [41] proposed stepwise regression model for

mediating effects to test the research hypothesis that citrus farming promotes the purchase of food from the market and enhances food nutritional levels. The specific mediation model constructed is as follows:

$$\begin{cases} Y_i = \theta_1 + cX_i + \varepsilon_1 \\ M_i = \theta_2 + aX_i + \varepsilon_2 \\ Y_i = \theta_3 + c'X_i + bM_i + \varepsilon_3 \end{cases} \tag{3}$$

where $Y_i$, $M_i$, and $X_i$ represent the dependent variable, mediator variable, and independent variables, respectively. Here, $M_i$ serves as a mediator variable, representing the extent to which households purchase food from the market. It is expressed as the share of food purchases made by the household from the market.

## 4. Results

### 4.1. Citrus Farming and Farm Production Diversity Impact on Dietary Diversity

Table 3 presents the regression estimation results, demonstrating the impact of citrus cultivation and farm crop diversity on household dietary diversity. The R-squared value and the F-statistic obtained from the model regression collectively signify the overall fit of the models. All the estimated outcomes withstand scrutiny based on dietary diversity indices at the food item level and crop diversity indices centered on citrus cultivation, indicating the robustness of the regression models.

**Table 3.** Estimation results of the basic model for household dietary diversity (measured as FCS).

| Variables | Model 1 | Model 2 | Model 3 |
|---|---|---|---|
| Citrus farming | 6.657 *** (1.112) | 3.344 ** (1.627) | 4.471 * (2.421) |
| Crop diversification | | 7.402 *** (1.368) | 6.302 *** (2.401) |
| Crop diversification × Citrus farming | | | 2.475 ** (1.124) |
| Age | | 0.039 (0.197) | 0.042 (0.197) |
| Edu | | 0.084 * (0.048) | 0.091 * (0.049) |
| Farmland | | 1.539 (1.139) | 1.557 (1.141) |
| Off-farm | | 0.188 (2.025) | 0.099 (2.036) |
| Com. food level | | 12.277 *** (2.614) | 12.179 *** (2.624) |
| Per capita exp. | | 6.803 *** (0.792) | 6.806 *** (0.793) |
| Family size | | 0.499 * (0.292) | 0.496 (0.330) |
| Ethnicity | | −0.444 (0.905) | −0.416 (0.908) |
| Fixed asset | | 7.263 *** (1.018) | 7.243 *** (1.020) |
| Constant | 56.65 *** (0.825) | 19.66 *** (4.082) | 21.04 *** (5.016) |
| $R^2$ | 0.107 | 0.345 | 0.401 |
| Prob > chi$^2$ | 0.000 | 0.000 | 0.000 |
| F-stat | 35.86 *** | 33.75 *** | 30.91 *** |

Note: standard errors are reported in parentheses. *, **, and *** indicate statistical significance at 10%, 5%, and 1% levels, respectively. Source: authors' survey.

In the regression results of Model 1, the estimated coefficient for citrus farming is found to be significantly positive, indicating a significant positive correlation between citrus cultivation and household dietary diversity. Our findings suggest that citrus farming plays a crucial role in reducing the risk of food insecurity by increasing food consumption expenditures and improving nutritional food intake, as indicated by consumption scores. However, in Models 2 and 3, which incorporate additional control variables and interaction effects, the significance of the regression coefficient for citrus farming diminishes. This implies that citrus farming's impact on household dietary diversity may be influenced by other variables. For instance, households engaged in citrus farming exhibit higher per capita household consumption (which corresponds to a higher income level) and a greater proportion of commercial food consumption. Hence, citrus farming is likely to induce changes in the Food Consumption Score (FCS) by elevating farmers' income and increasing

the purchase of food from the market. Model 2's regression results also demonstrate that variables such as household fixed assets, per capita expenditure, and the proportion of commercial food consumption significantly influence household dietary diversity. These findings are in line with the research of Kuma, Dereje, Hirvonen and Minten [13], Hashmiu, Agbenyega and Dawoe [15], who found that cultivating cash crops like coffee and cocoa in developing countries can enhance household income, stabilize consumption patterns, and, in turn, improve household dietary diversity for smallholder farmers.

The coefficient estimates for crop diversity indicators in Table 3 consistently indicate that both citrus-growing and non-citrus-growing households exhibit a positive correlation between farm crop diversity and household dietary diversity. The regression results from Model 2, considering other control variables, reveal that for every 10 percentage point increase in Shannon's Diversity Index of additional crops cultivated, household Food Consumption Scores (FCS) increase by 0.74 points. The results of the interaction effects in Model 3 demonstrate that, for citrus-growing households, crop diversity can be considered a significant factor explaining FCS, significantly enhancing dietary diversity in households where citrus is grown. This suggests that whether citrus cultivation directly or indirectly improves household food and nutritional security, households benefit from enhanced dietary diversity by practicing diversified production on the farm, including citrus, grains, and vegetables. This further underscores that in the Guangxi region of China, diversifying crop cultivation not only serves to mitigate agricultural risks [8] but also fulfills households' self-sufficiency in food supply. The findings presented above are in line with the research conducted by Islam, et al. [42], Ecker [24], and Bernzen, Mangnus and Sohns [29]. However, in contrast to the findings of Rajendran, Afari-Sefa, Shee, Bocher, Bekunda, dominick and Lukumay [16], the overall results suggest that crop diversification is indeed a strategy for enhancing household food security. However, it should be noted that increasing farm crop diversity does not necessarily lead to an increase in household income.

### 4.2. Citrus Farming and Farm Production Diversity Impact on Specific Food Groups

To conduct a more comprehensive examination of the specific dietary components through which citrus cultivation and crop diversity contribute to enhancing household dietary diversity, we employed regression analysis with the consumption scores for seven specific food categories serving as dependent variables. Notably, the category of oil and fat was excluded from the grouped regression analysis. This exclusion is primarily grounded in the prevailing dietary practices in China, where households incorporate edible oil into their daily culinary preparations. Consequently, this results in a negligible, discernible variance in household FCS attributable to oil and fat consumption. The regression outcomes across the three models align with the findings from the baseline regression analysis while also uncovering significant disparities among food categories. The estimations detailed in Table 4 consistently affirm that citrus farming markedly stimulates the consumption of dairy products, meat, eggs, fish, and fruits within households. Furthermore, the outcomes following the inclusion of control variables and interaction effects within the regression analysis align with the findings derived from the aforementioned baseline regression. Notably, household crop diversity exerts a pronounced influence on the consumption of staple grains, legumes, vegetables, and sugar, with this impact remaining statistically significant even after controlling for various control variables and interaction effects.

**Table 4.** Effects of citrus farming and crop diversity on food group consumption scores.

| | Cereals | Pulses and Nuts | Milk and Dairy | Meat, Eggs, and Fish | Vegetables | Fruits | Sugar |
|---|---|---|---|---|---|---|---|
| **Model 1** | | | | | | | |
| Citrus farming | 0.149 | −3.492 *** | 3.028 *** | 6.736 *** | −0.555 *** | 1.631 *** | 0.097 |
| | (0.112) | (0.589) | (0.405) | (0.742) | (0.126) | (0.195) | (0.087) |
| $R^2$ | 0.08 | 0.07 | 0.11 | 0.15 | 0.04 | 0.12 | 0.06 |
| F-stat | 38.84 *** | 35.07 *** | 55.81 *** | 62.48 | 19.23 *** | 69.37 *** | 10.26 *** |
| **Model 2** | | | | | | | |
| Citrus farming | 0.392 | −2.832 *** | 2.379 *** | 5.982 *** | −0.217 | 1.356 *** | 0.069 |
| | (0.257) | (0.653) | (0.517) | (0.889) | (0.151) | (0.252) | (0.113) |
| Crop diversification | 2.247 ** | 7.404 *** | −0.932 | −2.374 ** | 1.519 *** | 1.518 *** | 1.055 *** |
| | (1.124) | (0.792) | (0.628) | (1.078) | (0.183) | (0.306) | (0.137) |
| Control variables | Yes | Yes | Yes | Yes | Yes | Yes | Yes |
| $R^2$ | 0.31 | 0.33 | 0.21 | 0.29 | 0.22 | 0.16 | 0.12 |
| F-stat | 29.26 *** | 21.47 *** | 17.61 *** | 17.26 *** | 10.86 *** | 13.89 *** | 10.63 *** |
| **Model 3** | | | | | | | |
| Citrus farming | 0.291 | 0.727 | 0.294 | 0.127 (3.237) | −0.636 | 1.582 ** | 0.932 |
| | (0.478) | (2.382) | (1.891) | | (0.545) | (0.723) | (0.712) |
| Crop diversification | 3.235 *** | 9.493 *** | −2.155 * | −5.958 *** | 2.607 *** | 0.386 | 0.533 ** |
| | (0.313) | (1.559) | (1.238) | (2.119) | (0.357) | (0.604) | (0.270) |
| Crop diversification × Citrus farming | 1.326 *** | −2.804 | 1.642 | 4.812 ** | 1.459 *** | 0.178 | 0.789 ** |
| | (0.362) | (1.804) | (1.433) | (2.452) | (0.413) | (0.699) | (0.313) |
| Control variables | Yes | Yes | Yes | Yes | Yes | Yes | Yes |
| $R^2$ | 0.32 | 0.34 | 0.19 | 0.29 | 0.23 | 0.15 | 0.13 |
| F-stat | 28.65 *** | 19.94 *** | 17.09 *** | 16.24 *** | 13.24 *** | 11.22 *** | 10.12 *** |

Note: standard errors are reported in parentheses. *, **, and *** indicate statistical significance at 10%, 5%, and 1% levels, respectively. Source: authors' survey.

The significant and positively estimated coefficients presented in Table 4 concerning citrus farming's influence on meat products, dairy, and protein-rich food consumption signify that households engaged in citrus cultivation exhibit higher consumption levels of meat and dairy products compared to those not involved in citrus farming. It is important to note that meat and dairy products in Guangxi are typically procured through market transactions, underscoring the enhanced income and the concomitant contribution to commercial food consumption attributed to citrus farming. Simultaneously, the cultivation of a diversified array of crops serves as a direct facilitator for achieving household self-sufficiency in the consumption of legumes, fruits, and sugar, alongside other food items. This observation underscores that households in citrus-growing regions of Guangxi continue to maintain a substantial degree of self-reliance in terms of their food sources. It is worth highlighting that crop diversity may exert a dampening effect on the consumption of meat and dairy products within households. One plausible rationale for this phenomenon is that an increase in crop diversity significantly amplifies the consumption of non-meat food categories among households, consequently leading to a reduced intake of animal-sourced meat products. Furthermore, while crop diversification does contribute to stabilizing household income, it does not guarantee a commensurate increase in meat product consumption, given the inherent income uncertainty associated with this diversification [43].

Citrus cultivation does not exhibit a statistically significant effect on staple grain consumption in all three models. This lack of significance can potentially be attributed to the fact that staple grain consumption is a daily dietary category, and there is minimal variation among different households. This indirectly suggests that the people in Guangxi have significantly overcome poverty and food security issues in terms of grains [44]. Moreover, it is noteworthy that citrus farming does not exert a statistically significant influence on sugar consumption, potentially implying an indirect indication of a substitution effect between

citrus cultivation and sugarcane cultivation. This interpretation is underpinned by the prominence of Guangxi as a primary sugarcane-producing region in China, serving as a pivotal contributor to the nation's sugar production.

*4.3. Mediation Analysis of Citrus Farming Impact on Dietary Diversity through Market*

Building on the earlier research findings, citrus cultivation may improve households' dietary diversity by encouraging food purchases from the market. Consequently, this study validates its research hypothesis through a mediation model, examining whether citrus farming acts as a mediator in enhancing households' FCS through increased market food purchases. The results of the mediating effect are detailed in Table 5.

**Table 5.** Stepwise test results for the mediating effect of citrus farming on household dietary diversity.

|  | (1) | (2) | (3) |
|---|---|---|---|
|  | FCS | Com. Food Level | FCS |
| Citrus farming | 6.769 *** (1.210) | 0.278 *** (0.083) | 3.344 ** (1.627) |
| Com. food level |  |  | 12.277 *** (2.614) |
| Control variables | YES | YES | YES |
| $R^2$ | 0.358 | 0.252 | 0.345 |
| F-stat | 25.88 *** | 14.45 *** | 33.75 *** |

Note: standard errors are reported in parentheses. ** and *** indicate statistical significance at 5% and 1% levels, respectively. Source: authors' survey.

Regression (1) in Table 5 indicates that citrus farming has a significant direct effect on FCS, with an estimated coefficient of 6.769. Regression (2) shows that citrus farming significantly promotes households' purchase of food from the market, with an estimated coefficient of 0.278. In Regression (3), both citrus farming and the variable representing market food purchases are significant, indicating that even after controlling for the citrus farming variable, the mediating variable (Com. food level, share of all consumed food purchased on the market) still has a significant effect on households' FCS. As all three parameter estimates (coefficients a, b, and c in Equation (3)) are significant and ab and c have the same sign, it suggests the existence of a mediating effect from purchasing food in the market. The proportion of the total effect represented by the mediating effect is calculated as ab/c = 0.278 × 12.277/6.769 = 0.504. This implies that approximately 50% of the effect of citrus farming on households' FCS is achieved through the mediating role of purchasing food from the market. Therefore, the research hypothesis is validated, providing substantial evidence for the explanations and analyses discussed earlier in this study.

## 5. Discussion

The empirical findings of this study demonstrate that citrus farming not only directly enhances household dietary diversity but also aids in increasing household income, thereby assisting households in improving their dietary diversity by purchasing food from the market. Extensive research has already affirmed the income-generating effects of cultivating economic crops such as horticultural fruit trees [45,46]. However, there are also studies suggesting that planting economic crops may occupy land traditionally used for staple grain cultivation, potentially affecting household food security and increasing dietary diversity risks [47]. In the context of land scarcity in Guangxi, China, citrus cultivation may indeed result in a reduction in the land area dedicated to staple grain cultivation (Figure 1). Nonetheless, within the dual circulation model framework characterized by both domestic and international economic dynamics, facilitating enhanced income generation for households within resource-constrained landholdings may present a more favorable approach to ameliorating household food and nutritional security concerns in regions akin to Guangxi, China. With the rapid development of the market economy, market-based food purchases have emerged as the primary source of nutritional sustenance for farmers. It is important to note that our research does not advocate for farmers to

abandon staple grain cultivation in favor of citrus cultivation. Particularly in a densely populated country like China, achieving self-sufficiency in staple food security remains of paramount importance [3,48]. Moreover, our empirical research findings indicate that, whether cultivating citrus or not, crop diversity on farms contributes to the enhancement of household dietary diversity. Therefore, diversified crop cultivation remains an essential mechanism by which households can secure access to a broad spectrum of foods. Crop diversification affords farmers the opportunity for dietary variety, encompassing staples, vegetables, and fruits, and underscores its pivotal role in household nutrition and health, a facet that cannot be understated.

Crop diversity not only directly provides households with a wide array of food but can also influence household dietary diversity by stabilizing household income and altering household consumption patterns. The regression results in Table 3 provide indicative evidence of the primary pathways through which farm production diversity translates into household dietary diversity. Firstly, crop production diversity can directly increase dietary diversity by enhancing the household's ability to acquire a more diversified range of foods through more diversified food production [24,25]. The estimation results in Tables 2 and 3 demonstrate the production consumption and income effects in the regressions using food consumption score indicators and Shannon's crop diversity index. These results suggest that increasing household consumption expenditures, income-generating assets, and other household income variables do not alter the significance of the association between farm crop diversity and household dietary diversity. This implies a strong direct production–consumption effect. Secondly, crop diversity may also enhance household dietary diversity through changes in household income. Farm production diversity can increase farm income through means such as the cultivation of higher-priced cash crops, reduced income volatility, or improved soil fertility and ecological conditions, resulting in higher yields [8,22,25,49]. Finally, the cultivation of a variety of crops and the sale of diverse agricultural products can also potentially alter traditional household food consumption patterns [50], increase the channels through which households obtain food from the market [51], enhance awareness of nutritional and healthy food consumption [52], and, in turn, improve household dietary diversity. For example, households cultivating vegetables and fruits may have the opportunity to consume more vegetables, legumes, and fruits [24], while households engaged in livestock farming may consume more meat [53]. Moreover, the cultivation of diversified crops alongside the diversified sale of agricultural products can provide households with access to a greater variety of food market information and a more extensive range of food items from the market [54].

The empirical findings of this study also indicate that the impact of household education levels and non-agricultural employment on household dietary diversity is relatively weak, whereas the influence of income and asset levels is highly significant. It is generally believed that households with higher education levels possess a stronger awareness of food security, thus emphasizing family nutrition and health [55]. However, in developing countries, household education levels are often low and constrained by income, limiting the full potential of education. This underscores the necessity of providing essential food security and nutrition education to uplift households [56]. Furthermore, the increase in non-agricultural employment opportunities may not only directly augment income but also potentially introduce variations in household dietary diversity. The increased likelihood of eating outside the home due to expanded non-agricultural employment opportunities might reduce in-home dining, potentially leading to diversified effects on household dietary diversity [24,57,58]. Household income and wealth levels unquestionably represent key determinants of household food consumption. An increase in income levels naturally has the potential to elevate the level of food consumption and enhance dietary diversity [59,60].

While our research results do not directly advocate that citrus farmers should maintain a certain level of crop diversity, it is a concept worth thorough consideration by central and local governments, particularly in the context of China's comprehensive promotion of sustainable modern agriculture and rural revitalization. This consideration is especially

pertinent given the challenges posed by a large population of small-scale farmers in regions where land resources are scarce. The comparative analysis of food consumption scores related to crop diversity between citrus and non-citrus farming households in Guangxi further underscores the need to take into account the contributions of diversified crop cultivation to dietary diversity and food security. While the research findings may exhibit localized variations within emerging economies, their underlying inspirational value may apply to rural areas with similar agricultural and ecological conditions or at similar stages of development. In these regions, agricultural stakeholders share common characteristics as small-scale farmers and confront analogous food security concerns.

## 6. Conclusions

This study, based on a recent survey of citrus and non-citrus farmers in Guangxi, China, examined the impact of citrus farming and crop diversification on household dietary diversity. Multiple linear regression models, including a mediation effect model, revealed a significant role of citrus farming and crop diversification in enhancing household food and nutritional security, particularly among citrus growers. Citrus farming significantly influenced market-oriented food consumption, notably affecting meat and protein-based foods. Crop diversification consistently enhanced the Food Consumption Score (FCS) in both citrus and non-citrus cultivation households, promoting the consumption of staple foods, vegetables, fruits, and sugar. Non-citrus growers exhibited elevated self-sufficiency and reduced reliance on food markets, contrasting with citrus growers, who heavily depended on markets to enhance dietary diversity.

These findings emphasize the contributions of citrus cultivation and diversified crop production to food and nutritional security, with significant policy implications for sustainable agricultural development in western China. While maintaining high levels of crop diversification faces challenges during agricultural modernization, integrating horticultural industries with non-horticultural practices is proposed to improve food and nutritional security among smallholder farmers, particularly in horticulture-centric regions like Guangxi. However, this study has limitations, being region-specific and focused on citrus farmers, resulting in data with cross-sectional and geographical specificity. Future research should use panel data or a combination of quantitative and qualitative methods for more extensive investigations across diverse regions. The sustainable impact of smallholder crop diversity on agricultural development and biodiversity is evident, raising questions about sustaining crop diversity and enhancing food and nutritional security within the framework of agricultural modernization and warranting further research.

**Author Contributions:** Formulated and designed the research framework, X.C.; conducted the survey and analyzed the data, X.C., X.Z. and X.F.; provided the analytical tools used, X.C., X.Z. and B.Z.; manuscript—writing, X.C., B.Z. and X.Z. Subsequently revised multiple times by all authors. All authors have read and agreed to the published version of the manuscript.

**Funding:** This research was funded by the National Natural Sciences Foundation of China (grant No. 72063002) and the Natural Sciences Foundation of Guangxi Province (grant no. 2019JJA180063).

**Data Availability Statement:** The data presented in this study are available on request from the corresponding author. The data are not publicly available due to its inclusion of information that could compromise the privacy of research participants.

**Conflicts of Interest:** The authors declare no conflict of interest.

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
