# Peer review of "Produce or Buy: Impacts of Citrus Farming and Crop Diversification on Household Dietary Diversity in Guangxi, China"

_horticulturae, doi:10.3390/horticulturae9121256_

Round 1

Reviewer 1 Report

Comments and Suggestions for Authors

Dear Authors,

the topic chosen for your article seems really interesting even if it could lead to think that it is closely linked to China the analysis methodology adopted can easily be used and replicated everywhere. It is therefore potentially publishable in this magazine but, first, it requires a little further effort to make it truly captivating. In my opinion, in some passages of your manuscript it is better to simplify the text to make reading easier and more fluid.

Here are some suggestions that I hope will help improve your work:

-          Abstract: the part relating to the methodology is a little better. Please review the drafting of the abstract;

-          Keywords: do not repeat words that are already in the title;

-          Line 101: after  “:” the letter goes lowercase, so it must be “do” no “Do”;

-          Line 116-139: I think it would be better not to use personal expressions. I mean, in my opinion, it's better to say this work and not our work... and so on. Please check this part;

-          Line 110 and 139: check the editing of the points (.);

-          In the introduction I would focus much more on the "definition" of the research topic and less on the description of the study area (referring to the relevant paragraph for further information on the study area);

-          line 147: the bibliographic reference of the citation made is missing;

-          a map of the study area would greatly help a potential non-Chinese reader to identify the area investigated in this work;

-          move figure 1 into the text. In my opinion, it should be brought forward in order to have it close to the comment made on it;

-          it would be appropriate to also provide the economic values in dollars, as done for the hectares. Please check;

-          line 206: the source must be indicated as if it were a bibliographic reference in the text. Therefore it must be numbered in square brackets. This always applies;

-          paragraph 3.2: in my opinion it is necessary to provide (even if briefly) a better description of the FCS methodology so that even a non-expert reader can understand what we are talking about;

-          in table 1 only “**” is present. Then delete “*” from the note. Please check;

-          define and explain the elements that make up equation 1;

-          it would be appropriate to provide a “description of the characteristics of the interviewees”. Please add some descriptive statistics of the analyzed sample;

-          the fact that in model 3 the coefficients are no longer significant would require further investigation. It is not enough to say that perhaps other variables come into play. There is a need for a stronger and more consistent explanation (so one might say then you have the wrong model if you cannot explain the phenomenon). Please check;

-          it would be useful to provide some index of goodness of the regression models used (in addition to the R square);

-          check the spaces;

-          the conclusions lack the clarification of the limits of the work.

Good job!

Comments on the Quality of English Language

The English writing is quite good although some parts could be simplified a bit. Try to write in simple and concise English.

Author Response

Dear Reviewer,

Thank you for your valuable suggestions on our research manuscript. We have made revisions to the manuscript based on your comments. Please refer to the attached PDF response file for details.

Best regards

Reviewer 2 Report

Comments and Suggestions for Authors

Dear autnors

I am afraid the manuscript can not be read the way it is written in all the parts i.e. abstract: no methodology, hypothesis, nymerical results.

Introduction: to long without specific target to the research objective...

Methods: Where are the research hypotheses? are there models 1,2, etc below..... I didn't unerstand....

I am sorry but I wasn't able to read, and understand the resaerch and the results of your manuscript.

You should study previously published relative papers and rewrite the manuscript again in a clear understandable way.

Author Response

Dear Reviewer,

Thank you for your valuable suggestions on our research manuscript. We have made revisions to the manuscript based on your comments. Please refer to the attached PDF response file and revised manuscript for details.

Best regards

Reviewer 3 Report

Comments and Suggestions for Authors

Comments to the authors

This research utilizes newly gathered survey information from agricultural households in Guangxi, China. It differentiates between households involved in citrus cultivation and those that are not. The aim is to assess how citrus cultivation and crop diversification impact the food consumption of these respective households.

Abstract: Please revise the abstract to align with the author's guidelines, ensuring it does not exceed 200 words;

Page 2, line 69: Instead of possesses“ use possess“;

Page 2, line 81: Add a comma after fruits“;

Page 3, line 108: Instead of “remains“ use “remain“;

Page 3, line 110: After [33]“ use a full stop;

Page 3, line 114: Delete an extra space between [34],” and the word “rigorous”;

Page 3, line 115: Instead of “remain” use “remains”;

Page 3, line 125: Add “the” before “overall”;

Page 3, line 139: Add a full stop after “[20 ]”;

Page 3, lines 140-144: Please delete this paragraph;

Page 4, lines 174-175: “In 2022, the fruit yield of Guangxi province exceeded 31 million tons, while vegetable production exceeded 40 million tons (Figure 1).”. Please correct Figure 1, the text refers to the year 2022, and Figure 1 includes only 2021. In the corrected version of the Figure vertical axis should represent millions of tons, aligning with the text;

Page 5, line 211: Please adjust the chapter title “3. Data and Methods” to comply with the author's guidelines or instructions;

Page 5, line 219: Please add “the” before “household”;

Page 6, line 251: FCS abbreviation already introduced in the text;

Page 6, line 271: Please explain how the value of 56 is obtained;

Page 7, Table 1: Correct words in the first row: “Citrus farming” and “Non-Citrus farming” as follows “Citrus Farming Households” and “Non-Citrus farming Households”, respectively;

Page 8, Table 2: Define abbreviation “HH”. Delete “in” in three parentheses. Add space after “Household head”;

Page 8, line 308: Use only the abbreviation FCS;

Page 8, line 319, 320: Delete an extra space in these lines;

Page 9, Fussnote Table 3: Instead of “error” use a plural. The same is for Table 4;

Page 10, line 380: Add space before “Ecker…”;

Page 10, line 381: Add space before “But”;

Page 10, line 394: Use abbreviation instead Food Consumption Scores;

Page 13, line 483: Instead of “health” use “ healthy”;

Page 14, Conclusions: The conclusion needs to be more succinct and focused. Please revise to be more concise.

Please conform the references to adhere to the author's provided guidelines or instructions for formatting.

Comments on the Quality of English Language

The paper demonstrates an exceptionally high level of writing. Minor adjustments are highlighted within the provided comments.

Author Response

Dear Reviewer,   Thank you for your valuable suggestions on our research manuscript. We have made revisions to the manuscript followed your comments.  Please refer to the attached PDF response file for details.   Best regards

Round 2

Reviewer 2 Report

Comments and Suggestions for Authors

Dear authors

The manuscript can now be accepted, foolowing the revisions you made, and published 

Author Response

Thank you for all your comments and suggestions.

Reviewer 3 Report

Comments and Suggestions for Authors

The abstract still exceeds the word limit. Please condense it to a maximum of 200 words, as instructed;

The last paragraph in the Introduction has not been deleted;

The vertical axis in Figure 1 is not corrected;

I suggest condensing the conclusion and highlighting key points concisely;

The references do not adhere to the formatting guidelines specified for the respective journal.

Comments on the Quality of English Language

The proficiency in the English language is satisfactory.

Author Response

Thank you for all your comments and suggestions. We have revised the paper according to the specific comments in the main text.Please refer to the attached response document for specific modifications.
